theoretical biology, ecology, health and disease and epidemiology

vertical and horizontal transmission, environmental transmission, host–parasite dynamics, *Ophryocystis elektroscirrha*, monarch butterfly, *Danaus plexippus*

**Author for correspondence:**
Ania A. Majewska
e-mail: majewska.ania@gmail.com

# Multiple transmission routes sustain high prevalence of a virulent parasite in a butterfly host

Ania A. Majewska[1,2,4], Stuart Sims[1], Anna Schneider[5], Sonia Altizer[1,2] and Richard J. Hall[1,2,3]

[1]Odum School of Ecology, [2]Center for the Ecology of Infectious Disease, and [3]Department of Infectious Diseases, College of Veterinary Medicine, University of Georgia, Athens, GA, USA
[4]Department of Biology, Emory University, Atlanta, GA, USA
[5]Wisconsin Department of Natural Resources, Madison, WI, USA

 AAM, 0000-0002-0965-3177; SA, 0000-0001-9966-2773; RJH, 0000-0002-6102-4183

Understanding factors that allow highly virulent parasites to reach high infection prevalence in host populations is important for managing infection risks to human and wildlife health. Multiple transmission routes have been proposed as one mechanism by which virulent pathogens can achieve high prevalence, underscoring the need to investigate this hypothesis through an integrated modelling-empirical framework. Here, we examine a harmful specialist protozoan infecting monarch butterflies that commonly reaches high prevalence (50–100%) in resident populations. We integrate field and modelling work to show that a combination of three empirically-supported transmission routes (vertical, adult transfer and environmental transmission) can produce and sustain high infection prevalence in this system. Although horizontal transmission is necessary for parasite invasion, most new infections post-establishment arise from vertical transmission. Our study predicts that multiple transmission routes, coupled with high parasite virulence, can reduce resident host abundance by up to 50%, suggesting that the protozoan could contribute to declines of North American monarchs.

## 1. Introduction

Parasite virulence is often defined as the harm parasites cause their hosts, leading to reduced host fitness [1,2]. Optimal levels of virulence for parasites are thought to reflect a trade-off between the benefits of greater transmission (arising from within-host replication) and the costs of a shorter infectious period (owing to reduced host survival) [3,4]. Theory predicts that highly virulent pathogens should have more difficulty invading and persisting in host populations owing to the high mortality rate of infected hosts [5,6]. When outbreaks of highly virulent pathogens do occur, high infection prevalence tends to be followed by fade-outs that require introductions from sources external to the affected host population (or dormant infectious stages) to cause new outbreaks. Examples of harmful pathogens that cause outbreaks include phocine distemper virus in harbour seals [7] and plague in prairie dogs [8]. In contrast to pathogens associated with periodic outbreaks, those that maintain high prevalence in their hosts over time tend to cause low virulence, in some cases having no detectable survival costs in their primary hosts. Less virulent pathogens maintained at 50% or more infection prevalence include herpes simplex virus (HSV-1) in humans [9] and feline immunodeficiency virus (FIV) in lions [10]. As a result, a common assumption is that pathogens which infect a high proportion of hosts are relatively benign, and are therefore unimportant for regulating host abundance.

Empirical observations show that some highly virulent pathogens can maintain high prevalence in host populations [11–13]. This can occur in systems where human activity increases host densities; for example, over 50% of managed

European honeybee colonies suffer from the deformed wing virus, which shortens lifespan and causes deformities [12,14], and harmful sea lice in Pacific salmon near high-density salmon farms often exceed 50% [11,15]. In other cases, substantially virulent pathogens can persist at moderate prevalence by infecting multiple host species, or through the production of persistent environmental stages (e.g. rabies in the Serengeti, trichomoniasis in finches, [16,17]). Some parasites in arthropods with overlapping generations can also reach high prevalence, as exemplified by the sterilizing sexually transmitted *Coccipolippus* mite in ladybird beetles [18,19] and *Parobia* mites in eucalypt beetles that reduce host longevity and fecundity [20,21]. In pathogens for which virulence manifests as reduced fecundity rather than host mortality, virulence is less costly for parasite replication within hosts; a moderate parasite-induced reduction in fecundity may even be adaptive for sexual transmission by increasing mating opportunities for infected hosts [22].

Multiple transmission routes are a critical mechanism that could sustain high prevalence of virulent parasites—especially combined vertical (from parent to offspring) and horizontal (between unrelated hosts) transmission [23,24]. Seemingly paradoxically, vertically-transmitted parasites are expected to show low virulence, because transmission is closely tied to host reproduction [25,26]. However, general theory suggests that frequency- or density-dependent horizontal transmission, when added to vertical transmission, can allow parasite prevalence to increase to high levels, despite substantial mortality of the hosts [23,24]. Supporting evidence comes from the koala—*Chlamydia* system, where infections are transmitted both vertically and sexually, cause blindness in juveniles and urogenital infections in adults, and can affect up to 31% of wild koalas [27].

Monarch butterflies (*Danaus plexippus*) and the protozoan *Ophryocystis elektroscirrha* (*OE* hereafter) are well suited for exploring the dynamical consequences of multiple transmission routes for the spread and impacts of highly virulent parasites. Infections occur when caterpillars ingest spores scattered onto eggs or host plants by adult butterflies [28,29]. Parasites penetrate the gut wall and replicate internally; after host pupation, spores form around the scales of developing butterflies, and infected adults emerge covered with millions of dormant spores on the outsides of their bodies [30,31]. Infection can be debilitating or lethal for monarchs, causing reduced eclosion success, wing deformities, shortened adult lifespan and reduced flight performance [3,32]. Infection prevalence is low in monarchs that migrate seasonally, but reaches high levels (50–100%) in resident populations that breed year-round in the southern United States (US) and elsewhere [33,34].

Laboratory experiments and field studies have identified three transmission routes of *OE* in monarchs [28,29]. Vertical (*sensu* maternal) transmission occurs when an infected female sheds spores onto her eggs and surrounding leaves during oviposition, which are subsequently consumed by her offspring [30]. Past work showed that vertical transmission of *OE* approaches or exceeds 90% infection probability for offspring [28,29]. Second, environmental transmission occurs when infected adults scatter spores onto milkweed leaves which are later consumed by unrelated larvae [28,35]. Third, a relatively low number of spores can be transferred between adults during mating events or other contacts, and these adults can transmit spores to their offspring (hereafter adult transfer [28,29]). Monarchs that acquire spores as adults are temporary carriers, and themselves do not experience detrimental effects of the parasite [28].

Here we examined how multiple transmission routes allow a substantially virulent parasite to attain and persist at high prevalence, by pairing a field study with a mechanistic mathematical model parametrized for the monarch-*OE* system. We recorded an increase in infection prevalence in a wild monarch population, and quantified transmission in the field via environmental spore deposition and spore acquisition by uninfected adults. Our model explored how the relative contributions of three transmission routes change over time following parasite introduction, and determined which combinations of transmission routes facilitate high infection prevalence in hosts. We also predicted how changes in pathogen virulence and transmission can impact monarch population size.

## 2. Material and methods

### (a) Field study of parasite prevalence and transmission

#### (i) Sampling location

We sampled monarchs from a resident population in Savannah, GA, USA, at the Center for Research and Education at Wormsloe. We previously created six plots containing 15 tropical milkweeds (*Asclepias curassavica*) per plot within a 3 ha area (for site and plot details see [36]). The proximity of the garden plots to each other allowed free movement of adults between plots; therefore, for the purposes of this study, we consider the site as one milkweed patch. Past monitoring (2011–2014) showed that a high proportion of monarchs at this site and neighbouring locations were heavily infected with *OE* [33]. In January 2015, the site experienced a hard freeze that resulted in the dieback of milkweed and monarchs, subsequent growth of new milkweed, and re-colonization by monarchs in the spring. This provided the opportunity to quantify transmission dynamics following recolonization by the host and parasite, and to monitor changes in environmental, vertical and adult transfer transmission through time.

#### (ii) Monitoring infection prevalence

From May–October 2015, we tested 362 monarchs captured as adults for *OE* infection, following methods described in [37]. Briefly, we captured adults using aerial nets ($N = 7$–130 adults per month) and tested the adults for infection using non-destructive external sampling. Captures were made twice weekly from May to August, weekly in August, fortnightly in September, and once in October. Samples were examined at 60× magnification to quantify infection status. We classified adults with high spore loads (over 100 spores per sample) as heavily infected, indicating that infection was acquired during the larval stage. Samples with less than 100 spores commonly result from spore transfer between adult monarchs, and were classified as spore-contaminated (see below, [28]). We marked all adults with a small permanent ink number written on the hindwing to keep track of resampling of the same infected individuals.

We also measured *OE* infection in 350 monarchs sampled as larvae (and reared to the adult stage) to confirm that prevalence reflected acquisition of infection at the site (rather than immigration of infected adults from other locations). We monitored milkweed plants once a week and collected a subset of late instar larvae (that would have been exposed to infection prior to collection) to rear individually to the adult stage in

the laboratory ($N = 11$–$201$ larvae month$^{-1}$). Following adult eclosion, we recorded the sex, marked, and tested adults for infection, and released reared monarchs at the study site within 7 days of emergence. We asked whether the probability of infection for monarchs was predicted by sex, stage of capture (adult versus larva), month (linear and quadratic terms to account for nonlinearity), and the interaction between stage and month. We assessed differences in infection probability via logistic regression models using R version 3.6.0 [38].

### (iii) Measuring horizontal transmission

From June–September 2015, we marked and released 163 uninfected adult monarchs to quantify adult transfer (spores acquired by adult monarchs through mating or other activity). Uninfected monarchs were those that we collected as late instar larvae ($N = 110$), or that we captured in the plots as previously unmarked uninfected adults ($N = 53$). We resampled all recaptures ($N = 7$–$26$ monarchs recaptured month$^{-1}$) to quantify the proportion of previously uninfected monarchs that acquired at least one $OE$ spore via contacts with infected adults, and to count acquired spores per individual.

To estimate environmental transmission via consumption of spores deposited onto milkweed, we randomly chose tropical milkweed plants from each plot and sampled milkweed leaves by feeding cuttings of the plants to uninfected larvae. Monarch larvae were outcrossed descendants of wild uninfected autumn migrants (collected from St Marks, FL, USA in October 2014) that overwintered in the laboratory as adults. Each month from May to September, we took cuttings (top half portion of the stalks) from five randomly chosen tropical milkweed plants per plot, removed any eggs and larvae, and placed stalks in 0.5 l plastic containers with mesh screen lids. A single uninfected first instar larva was added to each container ($N = 2$–$3$ containers plant$^{-1}$ month$^{-1}$, for a total of 376 exposed larvae that survived to adulthood). We chose first instar larvae to provide an intermediate estimate of infection probability. Prior experiments showed that for a given spore dose, first instar is moderately susceptible to $OE$, second instar is most susceptible, and past third instar susceptibility decreases sharply [31,39]. Once monarchs reached the fifth instar (least susceptible to infection), they were fed greenhouse grown (uncontaminated) milkweed. Following eclosion, adults were tested for infection as described above. For each month, we calculated the proportion of plants for which at least one larva emerged as an infected adult.

## (b) Model development

We modified an existing stage-structured differential equation model of the monarch-$OE$ interaction [35] to examine the dynamical consequences of multiple transmission routes for parasite prevalence. The model tracks the number of susceptible and infected monarch larvae ($S_L$, $I_L$ respectively) and adults ($S_A$, $I_A$), as well as the total number of milkweed leaves ($M$) and the number of leaves receiving an infectious dose of protozoan spores ($M_C$). We extended this model to incorporate adult spore transfer by tracking an additional class of uninfected adults that acquire $OE$ spores via mating and other contacts with infected adults (designated by $C_A$), and explicitly accounted for the growth and consumption of milkweed. The model schematic and equations are provided in figure 1, with parameters described in the electronic supplementary material, table S4.

### (i) Monarch and milkweed population dynamics

In the absence of infection, we assume that the per adult egg production rate is $b_S$; eggs hatch into larvae after a development time of $\tau_e$. Larvae experience natural mortality at per capita rate $\mu_0$, and additional mortality at a rate proportional to larval density per milkweed (with scaling parameter $\mu_d$). Surviving larvae develop to pupae at rate $g$, and emerge as adults after $\tau_p$ days in the pupal stage (figure 1, equations (2.1) and (2.3)). We assume a 50 : 50 sex ratio, and adult mating shortly after eclosion; adults die at per capita rate $\mu_S$ (figure 1, equation (2.3)). Milkweed leaf biomass is assumed to grow logistically (with maximum growth rate $r$ and carrying capacity $K$) and is lost via larval consumption (at per capita rate $c$; figure 1, equation (2.6)).

### (ii) Parasite transmission and costs of infection

Because vertical transmission is imperfect, a fraction $p_v$ of progeny from infected females become infected, with the remaining $1 - p_v$ offspring escaping infection (figure 1, equations (2.2) and larval production). Adult spore transfer most commonly occurs through mating of uninfected and infected adults (at rate $\delta$) and occurs both ways between the sexes [29]. However, our model assumes that the subsequent transfer of spores to offspring occurs from females only. Females can infect a high proportion of their offspring within several days after mating with an infected male [28], while males would need to mate with a healthy female shortly after acquiring spores through adult transfer, and the rate of larval infection would be minimal. Spore-contaminated females produce eggs at per capita rate $b_S$, and a fraction $p_h$ of their offspring become infected (figure 1, equations (2.4) and larval production). Monarchs that experience adult transfer lose spores at rate $\mu_C$ and return to the susceptible class. Environmental transmission occurs when infected adults visit milkweed leaves and deposit an infectious spore dose (at per capita rate $\lambda$; figure 1, equation (2.7)). Larvae become infected by ingesting these leaves based on their consumption rate ($c$) and the relative frequency of contaminated leaves ($M_C/M$). Contaminated leaves revert to a non-infectious state at rate $\mu_w$.

Prior work [3] showed that monarchs infected by $OE$ as larvae have a lower probability of eclosing and finding mates ($\theta$), produce fewer eggs per day (at per capita rate $b_I$) and experience higher adult mortality ($\mu_I$; figure 1, equation (2.5)). To examine how model outcomes respond to individual-level impacts of infection, we developed a composite measure of virulence, defined as the proportional reduction in lifetime reproductive success relative to uninfected adults. Briefly, this measure incorporates infection-induced reductions in adult lifespan, egg production, and the probability of eclosion and mating, relative to parameter estimates for uninfected adults (see the electronic supplementary material for derivation). Based on empirically-informed parameters (electronic supplementary material, table S4), we estimated that infected monarchs on average experience a 58% reduction in fitness compared to uninfected adults. We explored the effects of changes to pathogen virulence by varying the infected adult mortality rate ($\mu_I$) and holding other components of virulence constant, across a range corresponding to relatively low virulence (40% reduction in fitness) to very high virulence (100% reduction in fitness).

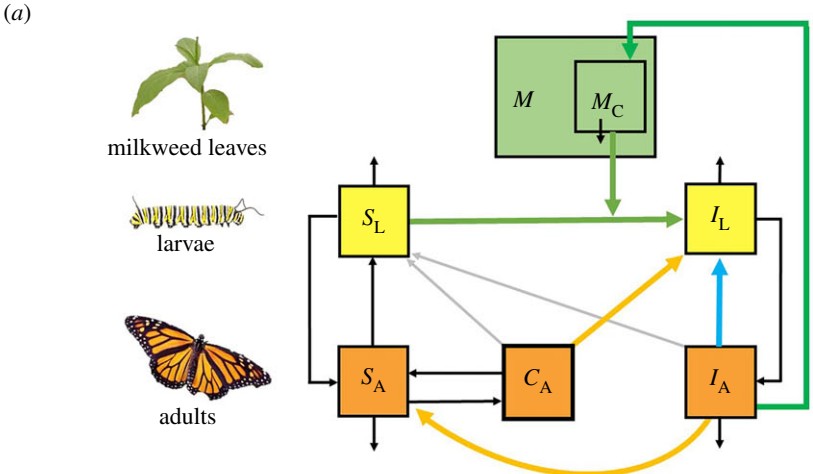

(a)

milkweed leaves

larvae

adults

(b)

**larval dynamics**

$$\frac{dS_L}{dt} = B_S(t - \tau_e) - \left(\mu_0 + \mu_d \frac{N_L}{M}\right)S_L - gS_L - c\frac{M_C}{M}S_L \tag{2.1}$$

$$\frac{dI_L}{dt} = B_I(t - \tau_e) - \left(\mu_0 + \mu_d \frac{N_L}{M}\right)I_L - gI_L + c\frac{M_C}{M}I_L \tag{2.2}$$

larval production $\quad B_S = b_S S_A + b_I(1 - p_v)I_A + b_S(1 - p_h)C_A$

$$B_I = b_I p_v I_A + b_S p_h C_A$$

**adult dynamics**

$$\frac{dS_A}{dt} = g\rho S_L(t - \tau_p) - \mu_S S_A - \delta S_A\left(\frac{I_A}{N_A}\right) + \mu_C C_A \tag{2.3}$$

$$\frac{dC_A}{dt} = \delta S_A\left(\frac{I_A}{N_A}\right) - \mu_S C_A - \mu_C C_A \tag{2.4}$$

$$\frac{dI_A}{dt} = \theta g\rho I_L(t - \tau_p) - \mu_I I_A \tag{2.5}$$

**milkweed dynamics**

$$\frac{dM}{dt} = rM\left(1 - \frac{M}{K}\right) - cN_L \tag{2.6}$$

$$\frac{dM_C}{dt} = \lambda\left(1 - \frac{M_C}{M}\right)I_A - c\frac{M_C}{M}N_L - \mu_w M_C \tag{2.7}$$

**Figure 1.** (a) Diagram of the monarch-*OE* compartment model, with hosts subdivided by life stage and infection status ($S_L$ = abundance of susceptible larvae, $S_A$ = abundance of susceptible adult monarchs, and $C_A$ is the abundance of previously uninfected adults that become contaminated with *OE* spores via adult transfer). Infected larval and adult abundances are represented by $I_L$, and $I_A$, respectively. Milkweed leaves (*M*) can become contaminated ($M_C$) with spores when infected adults deposit an infectious spore dose. The coloured arrows represent the three transmission routes examined: vertical transmission from infected females to offspring (blue line), transmission to larvae by spores acquired during the adult stage (i.e. adult transfer, orange line), and environmental transmission where spores shed onto milkweed by adults are ingested by unrelated larvae (green line). (b) Equations of the stage-structured compartment model representing parasite transmission in monarchs. Yellow box tracks larval dynamics ($S_L$, $I_L$), orange box tracks adult dynamics ($S_A$, $C_A$, $I_A$) and green box tracks milkweed dynamics (*M*, $M_C$). $N_L$ and $N_A$ are respectively the total number of larvae and adults. Parameter definitions and reference values are provided in the electronic supplementary material, table S4. (Online version in colour.)

### (iii) Model parametrization and analysis

We parametrized our model for a typical tropical milkweed resident breeding site in the southeastern US, using field observations of the Savannah, GA site, and previously published work (for derivations and parameter list see the electronic supplementary material). We initiated the model with 18 uninfected and two infected adults, tracked the number of new infections arising from each transmission

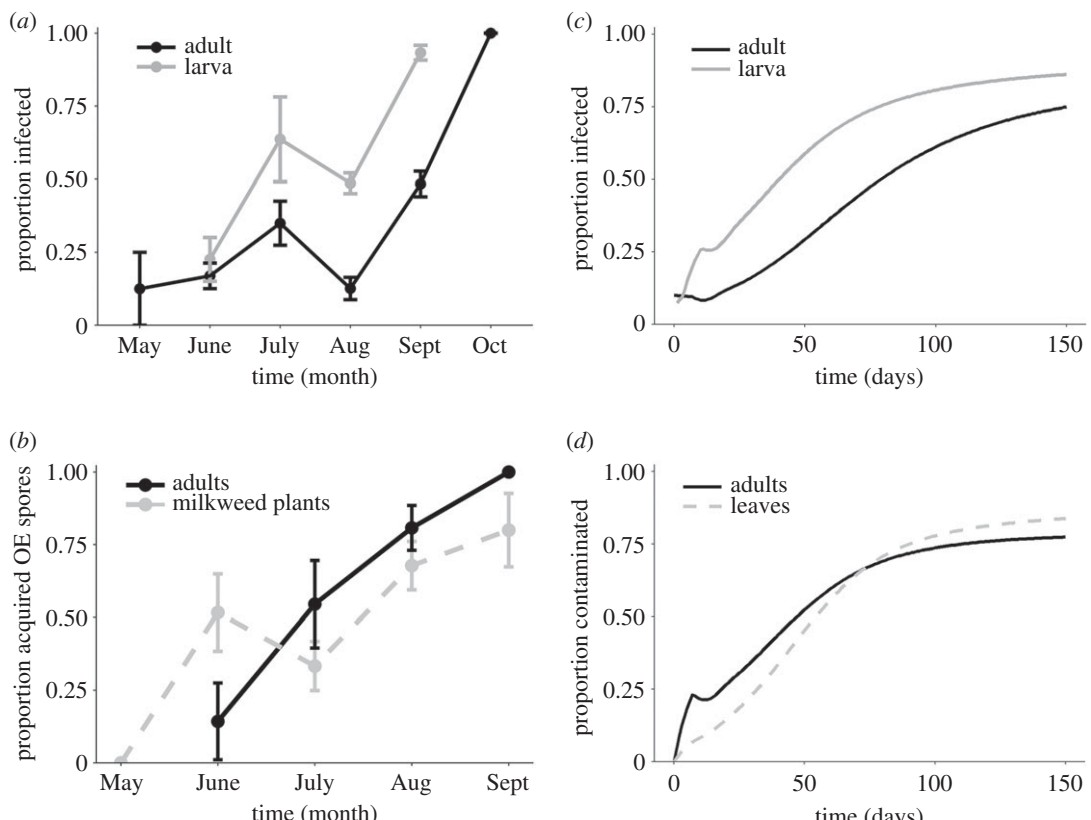

**Figure 2.** Observed (L) and model-predicted (R) changes in monarch infection prevalence. (*a*) Field observation of the proportion of heavily infected wild monarchs collected as adults (black line; $N = 7$–130 month$^{-1}$) and larvae (grey line, $N = 11$–201 month$^{-1}$). The average infection prevalence was higher for larvae ($0.57 \pm 0.03$; $N = 350$) relative to adults ($0.38 \pm 0.03$; $N = 362$). (*b*) Proportion of wild adults (solid black line; $N = 7$–26 month$^{-1}$) and milkweed plants (grey dashed line; $N = 23$–31 month$^{-1}$) that acquired spores over time. (*c*) Model predictions of the proportion of infected adults (black line) and larvae (grey line) over time. (*d*) Predicted proportion of susceptible adults (black line) and milkweed leaves (grey dashed line) that become contaminated with *OE* spores ($C_A/(C_A + S_A)$ and $M_C/M$ respectively). We initiated the model with two infected and 18 healthy adults, and parameter values are as presented in the electronic supplementary material, table S4.

route through time, and assessed the importance of each route in determining parasite prevalence by numerically solving variants of the model with different transmission modes switched off. We ran the model for 150 days, corresponding to the duration of our field observations from May–September (and before autumn migrants pass through the site in October–November). We also explored how different transmission routes interact with parasite virulence to determine parasite-mediated impacts on host population size. We used R package dESolve [38,40] to solve the system of differential equations.

## 3. Results

### (a) Field data on infection prevalence, environmental transmission and adult transfer

The proportion of infected monarchs sampled in field plots increased from low levels to close to 100% during the observation period (May–October; $z = 6.60$, $p < 0.001$), with similar rates of increase observed for monarchs collected at either the adult or larval stage (figure 2*a*). Of the 163 uninfected adults released into field plots, we observed 70 the following week or later, of which we recaptured 52 to test for parasite spores. In total, 81% of adults acquired *OE* spores, with a mean of $90.5 \pm 47.7$ spores per positive adult ($N = 42$). The probability of spore acquisition and number of spores acquired increased over time (figure 2*b*; $z = 3.78$ and $z = 4.40$, $p < 0.001$, respectively).

Most captive-reared larvae fed tropical milkweed stalks collected from the plots survived to adulthood (82%), and of these, 33% ($N = 308$) emerged as infected adults. The proportion of milkweed plants for which at least one monarch emerged as an infected adult increased over time, from zero to approximately 75% ($z = 5.84$, $p < 0.001$; figure 2*b*; electronic supplementary material, table S3).

### (b) Model analysis of transmission modes

Numerical solution of the transmission model showed that the proportions of infected larvae and adults increased rapidly following initial colonization, and reached high levels (approximately 75%, figure 2*c*) by the end of the 150-day period. Predicted dynamics and late season prevalence were similar to observed field infection prevalence in the current study (figure 2*a* versus 2*c*), and past observations in resident (non-migratory) monarchs [33,37]. The proportion of uninfected adults and milkweed leaves that acquired *OE* spores also increased through time, each reaching approximately 70% by the end of the simulation period (figure 2*d*).

Model exploration showed that the contributions of each transmission route to the proportion of new infections changed over time. Early in the dynamics (ignoring first few days, which represent small numbers of infected individuals), the two horizontal transmission routes (environmental and adult transfer) cause most new infections (figure 3*a*). However, once the parasite became established and more than half of the hosts were infected, vertical transmission caused the majority of new infections (figure 3*a*). To understand

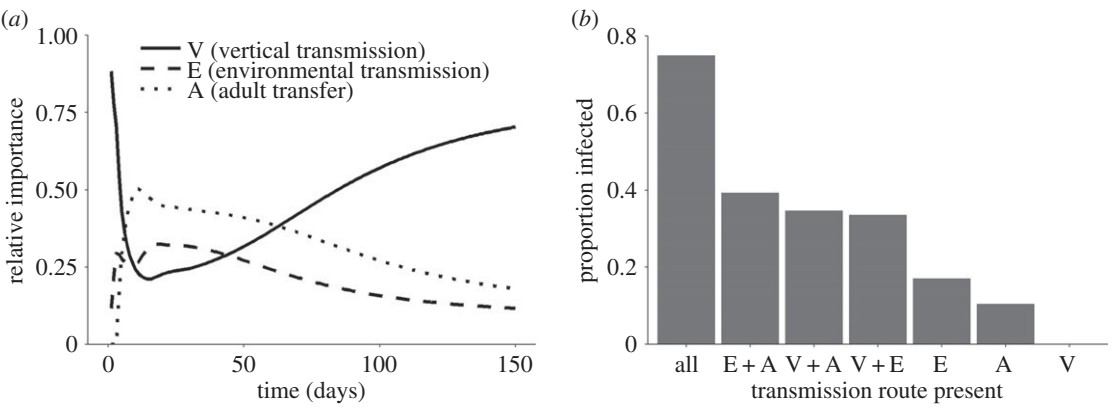

**Figure 3.** (*a*) Changes in the contribution of three transmission routes through time since initial colonization by monarchs as predicted by the model. Lines show the proportion of all new cases attributed to each transmission route. Early in the dynamics, environmental transmission (dotted line) and contamination of previously uninfected adults via adult transfer (dashed line) contribute to most new infections. Once the parasite becomes established, most new cases are caused by vertical transmission (solid line). (*b*) Predicted final infection prevalence after 150 days (based on adults infected as larvae) given different combinations of transmission routes.

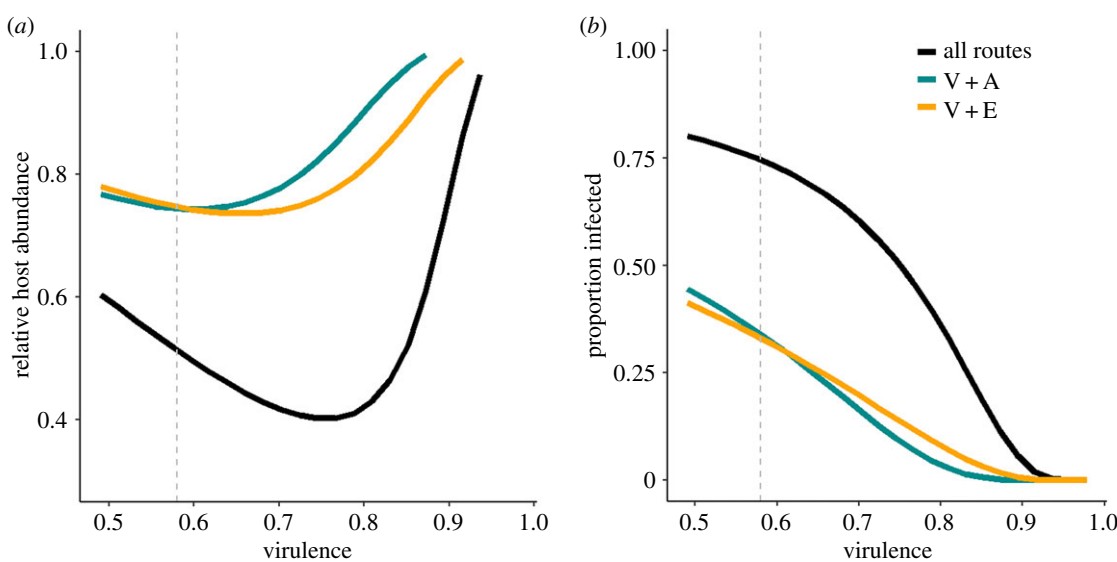

**Figure 4.** (*a*) Model-predicted abundance of adults 150 days post-colonization relative to a parasite-free population and (*b*) proportion of infected monarchs as a function of virulence (the proportional reduction in lifetime reproductive success relative to uninfected adults; see the electronic supplementary material for derivation). Dashed vertical line indicates virulence of 0.58, estimated from our parametrization. Coloured lines reflect different transmission scenarios as follows: black line, all three transmission routes; blue line, vertical and environmental transmission only; orange line, vertical and adult transfer only. (Online version in colour.)

the relative importance of the three transmission routes to overall prevalence at the end of the simulation period, we quantified the final proportion of infected monarchs (ignoring contaminated adults) using all seven possible combinations of the three transmission routes (i.e. each transmission route operating alone, in pairwise combinations, and all three operating together). The parasite was unable to establish with vertical transmission alone. Environmental transmission and adult transfer operating singly, and pairwise combinations of transmission routes, resulted in relatively low infection prevalence (less than 40%; figure 3*b*). By contrast, the presence of all three routes operating simultaneously produced infection prevalence greater than 50% (figure 3*b*), and this outcome was observed across a range of other parameter values (electronic supplementary material, figures S1–S3).

We explored how parasite virulence interacts with transmission to reduce host population size, defined as the end of season adult monarch abundance, scaled by the population size in the absence of infection (figure 4*a*). Virulence

was varied in the model by changing the infected adult mortality rate across values corresponding to a 40–100% reduction in host lifetime reproductive success following infection. Host population size showed a U-shaped relationship with virulence for pairwise combinations of transmission routes, and for all three routes operating simultaneously (figure 4*a*). Parasites that reduced host fitness by 50–80% persisted at high prevalence (figure 4*b*), infected half or more of the host population, and caused declines in host abundance of 50% or more when all three transmission modes operated simultaneously. Combinations of two transmission modes predicted less severe population impacts, relative to the three-transmission mode scenario (figure 4).

We explored the sensitivity of model outcomes to variation in the parameters and performed Latin Hypercube Sampling across all combinations of parameter values to determine the impact on late season prevalence (see the electronic supplementary material, figures S1–S3 and for additional details). Increasing host fecundity had a strong negative effect on prevalence. There was no net directional

effect of density-independent larval survival on prevalence, but increasing larval density tended to increase prevalence. For environmental transmission and adult transfer, increasing the transmission rate or duration of spore contamination tended to increase prevalence to a larger degree than changes to larval density, with the most pronounced increase in response to the probability of spore-contaminated females infecting their offspring.

## 4. Discussion

Our work showed that a substantially virulent vertically transmitted parasite can persist at high prevalence, and substantially reduce host population size, provided that multiple routes of horizontal transmission also occur. Simultaneous transmission via vertical, environmental and adult transfer routes predicted high prevalence of the protozoan *OE* in monarchs, consistent with empirically based prevalence estimates from this study and prior field monitoring in resident monarch populations (e.g. [33]). Model analysis suggested that the importance of each transmission route changes over time following host and parasite colonization. Early horizontal transmission is crucial for parasite invasion because initially infected adults are rare (and thus, so is vertical transmission from females to offspring); spore deposition onto milkweeds and spore transfer to uninfected adults greatly increases the pool of susceptible larvae to infect. However, once the majority of adults are infected, horizontal transmission is outpaced by vertical transmission, because most females can infect a very high fraction of their offspring. These findings advance understanding of the mechanisms that maintain highly virulent parasites at high prevalence in host populations, with consequences for predicted impacts on host population size.

Previous work on *OE* infection in resident monarch populations showed consistently high (50–100%) prevalence in the southern coastal US [33,37]. Resident monarchs are predominately associated with planting of the non-native tropical milkweed (*A. curassavica*) in gardens and parks [34]. Unlike most milkweeds native to eastern North America, tropical milkweed does not senesce during autumn, and can provide food year-round for larval and adult monarchs in warm climates [41,42]. Past work showed that migratory monarchs that breed seasonally on native milkweeds have low infection prevalence, probably through migratory mechanisms that reduce parasitism [43,44]. However, even in the absence of these mechanisms, explaining high infection prevalence in resident monarchs has remained challenging owing to the debilitating nature of infection, which should otherwise lower the fitness of infected monarchs and limit parasite transmission [28,35].

Field monitoring following site recolonization allowed us to compare infection increases in the field to predicted model dynamics. In both observed and predicted dynamics, *OE* prevalence increased rapidly and reached 75–100% prevalence by October. We found strong evidence for environmental transmission via parasite spore deposition on milkweed, with most field-collected milkweed acquiring *OE* spores over several months. Previous work indicated that these spores can remain viable on milkweeds for many weeks [28,35]. We also found a high rate of adult spore transfer, with both observed data and model output indicating that 80–100% of previously uninfected adults acquire low

numbers of spores once *OE* prevalence increases to high levels. Past studies suggest that most adult contamination occurs during mating [28], which involves male : female struggles that last for up to 30 min; if males are successful, the pair remains coupled for up to 16 h [45]. However, it is important to note that up to a third of all mating attempts are male : male, and that other contacts, such as territorial defence contacts by males towards other individuals might contribute to spore transfer between adults. Adult spore transfer could be particularly important for parasites to persist in migratory monarch populations, given that healthy adults live longer and show greater flight performance than infected monarchs [32,46].

The monarch-parasite interaction examined here underscores the importance of horizontal transmission to the short-term dynamics and persistence of a highly virulent parasite with clearly-known vertical transmission. In agreement with previous general theoretical studies [23,24], our model showed that either of the two horizontal transmission routes can permit the pathogen to invade and persist, relative to the case of vertical transmission alone, which could not support pathogen invasion. All three transmission routes operating simultaneously are needed to produce the very high prevalence observed in resident monarchs. Another recent model of monarch-*OE* dynamics, which included only vertical and environmental transmission, was able to capture observed dynamics at northern (seasonal) breeding sites, where prevalence reaches 10–20% by August [35]. However, even assumptions of extremely high spore deposition on plants, and spore longevity for many months, could not produce infection prevalence above 15%, suggesting that the transfer of low numbers of spores to previously uninfected adults is a crucial transmission route in this system.

A growing number of studies, including ours, suggest that even seemingly minor transmission routes can be crucial for understanding, predicting and managing pathogens [47–49]. For example, work on the Ebola outbreak showed that the virus spreads not only via direct physical contact but also sexually, and incorporating both transmission routes in a model led to drastically different predictions for infection dynamics [50,51]. The addition of alternative transmission routes can be critical for determining pathogen invasion thresholds [52]. For instance, adding environmental transmission allowed avian influenza virus to persist in wild waterfowl populations and increased the risk of epidemic resurgence [53]. Finally, anther-smut fungal infections of the alpine carnation are typically transmitted by insect pollinators as vectors, but recent work showed that the previously understudied transport of spores by wind can explain the observed high prevalence of this sterilizing disease [54].

The high prevalence maintained by multiple transmission routes in the monarch-*OE* system can contribute to substantial host population declines. We estimate that *OE* in resident monarch populations reduces monarch population size by about half, relative to population size expected in a parasite-free state. Interestingly, other vertically transmitted pathogens, such as the *Wolbachia* bacterium in *Hypolimnas* butterflies, can persist at high prevalence and substantially lower host population size by killing male embryos [55]. In monarchs, increased *OE* virulence can intensify this loss, suggesting that the introduction of a more virulent strain could threaten the viability of local populations. At the same time, empirical work indicates that virulence could be

buffered by the medicinal properties of tropical milkweed that reduce within-host infection load and increase adult lifespan [56]. Although our model parametrization did not account for changes in parasite virulence caused by tropical milkweed, we predict that reduced virulence alone might increase infection prevalence, and thus could exacerbate, not ameliorate, parasite infections at the population level. Further theoretical work is needed to explore the impact of medicinal plant use on monarch-parasite dynamics at local and regional scales.

Migratory monarchs in eastern North America have undergone recent declines in overwintering numbers (an 84% reduction from 1996–2017). This decline is thought to be caused by multiple factors, including habitat loss throughout the annual migratory cycle [57,58]. Citizen science [59] and historical data [33] suggest that tropical milkweed and year-round monarch breeding have become common in recent decades, potentially linked to warmer winters and planting of milkweeds intended to help monarchs. Further shifts from migratory to resident status of North American monarchs will probably favour increased transmission and population-level impacts of *OE* parasites. Thus, monarchs can be added to the growing number of case studies indicating that human activities are intensifying pathogen pressures on wildlife [60]. More work is needed to examine how practices in urban backyards, gardens and parks, such as providing novel and predictable food resources, might impact wild host–pathogen dynamics (e.g. [61]).

While the presence of multiple transmission routes is one mechanism that can support persistence of a highly virulent pathogen, complex spatial dynamics, imperfect immunity, timing of host life history events, and alternative host species can also allow very virulent pathogens to persist [19,62–64]. For monarchs, milkweed distribution might differ across the southern US compared to more northern latitudes, where native milkweeds are common in open fields and roadsides [65]. In the southern US, tropical milkweed plants are typically associated with gardens and parks, resulting in patches of high densities of tropical milkweed across the landscape, which might crowd monarchs and increase transmission rates. Future studies could explore how adult monarch movement between habitat patches, and heterogeneity in monarch density and milkweed plants, influences transmission dynamics in this system.

In closing, our study suggests that multiple transmission routes support high infection prevalence of a debilitating parasite in wild monarch populations. More broadly, omission of alternative transmission routes in modelling infection dynamics can significantly underestimate outbreak size and pathogen impacts on host abundance. This finding is pertinent to the migratory monarchs of eastern and western North America which have experienced notable declines in recent decades [66,67]. Although parasite infection has not been raised as a significant threat to monarch population viability, our analysis suggests that *OE* might warrant future consideration, as it could substantially reduce local populations. Our results are also relevant for other pathogens present at high levels in wild hosts, which are commonly assumed to cause little harm (e.g. FIV in lions, [10]), yet might significantly reduce host abundance under scenarios of moderate to high virulence.

**Data accessibility.** Data along with R code are available from the Dryad Digital Repository: https://doi.org/10.5061/dryad.628pd5n [68] and github: https://github.com/Majewska/Majewska_et_al2019_MonarchOEmodel.

**Authors' contributions.** A.A.M., R.J.H. and S.A. designed the study; A.A.M., S.S. and A.S. collected field data; A.A.M. conducted the statistical analyses; A.A.M. and R.J.H. developed and analysed mathematical models. A.A.M. wrote the manuscript with input and revisions from all authors.

**Competing interests.** We declare we have no competing interests.

**Funding.** This work was supported by the Wormsloe Foundation fellowship and grants from Odum School of Ecology of The University of Georgia, Joan Mosenthal DeWind Award of the Xerces Society, Pollinator Partnership, and Monarch Joint Venture awarded to A.A.M.; A.S. was supported by the Population Biology of Infectious Diseases NSF REU [DBI-1156707] which also provided financial support for this work; S.A. and R.J.H. were supported by the NSF [DEB-1754392] and S.A. by SERDP-RC2700.

**Acknowledgements.** We thank Craig and Diana Barrow providing field plot locations, and Sarah Ross for financial and logistical support. We thank Andy Davis, Hunter Anderson and Christopher A. Cleveland for support in establishing and maintaining plots. We also thank Reidar Crosswell for assistance in field data collection. We are grateful to the members of the Altizer laboratory for comments on previous drafts of the manuscript and to the funding agencies whose support was essential to our work.

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
