## [Reviewer comments · Proceedings of the Royal Society B: Biological Sciences]

Review History

RSPB-2019-0343.R0 (Original submission)

Review form: Reviewer 1 (Emme Bruns)

Recommendation

Accept with minor revision (please list in comments)

Scientific importance: Is the manuscript an original and important contribution to its field?

Excellent

General interest: Is the paper of sufficient general interest?

Excellent

Quality of the paper: Is the overall quality of the paper suitable?

Excellent

Is the length of the paper justified?

Yes

Should the paper be seen by a specialist statistical reviewer?

No

Do you have any concerns about statistical analyses in this paper? If so, please specify them explicitly in your report.

No

It is a condition of publication that authors make their supporting data, code and materials available - either as supplementary material or hosted in an external repository. Please rate, if applicable, the supporting data on the following criteria.

Is it accessible?

N/A

Is it clear?

N/A

Is it adequate?

N/A

Do you have any ethical concerns with this paper?

No

Comments to the Author

The authors used a combination of new field studies and published data to quantify three different transmission modes of a virulent parasite onto a population of non-migrating monarch butterflies. They developed a compartmental model, parametrized with field data, to estimate the impact of these three transmission modes on disease dynamics. They found that all three modes were needed to explain the high level of disease prevalence observed in the field, and that the contribution of these modes differed over time.

I really enjoyed reading this paper. Transmission is at the heart of disease dynamics and yet, we still know much less about transmission in natural systems than virulence. The study makes the important point the seemingly trivial transmission routes (adult -transfer) can nevertheless play an critical role in disease establishment and spread. It also nicely illustrates the point that multiple transmission routes can lead to high sustained prevalence, even for a highly virulent disease. I found the manuscript to be well written overall, but I have a few questions and points of clarification.

Point of clarification: It is mentioned that there are 6 plots. Are these close enough that the dynamics are the same? (Is the model for the whole patch?)

Out of curiosity, how does one mark adult monarchs?

How were the milkweed leaves sampled? Were these always the top leaves? What proportion of the leaves had larvae on them to begin with?

Fig 1 and table 4: The Density-dependent per capita larval mortality rate (u_1) and mortality rate of infected adults (u_I) look very similar.

Page 11 first paragraph: "The stage of collection, and the interaction between stage and month, were not significant predictors of infection prevalence" –what does 'stage of collection' mean?

Does it mean adult vs larvae, or larval stages (because larvae v. adult looks like a big effect)?

The line "Adult transfer of spores most commonly occurs through mating of uninfected females

to infected males (at rate \square)..” on page 9 seems surprising, because sex-specific differences in transmission have not been mentioned yet. I would suggest citing the source in the text (I know it is in table S4) or mentioning this bit of natural history in the introduction.

It is a bit unclear to me how the ‘composite virulence measure’ (described on pages 9 & 10) is incorporated into the model. It looks like the the individual components of virulence are incorporated into eq 2, 3 (which makes sense). Is this composite measure just for the analysis shown in Fig 4? If so, you might tweak the wording in the methods to clarify.

Related: page 13, top paragraph. Point of clarification: monarch population size is determine through the model output, and virulence is that composite measure reference earlier, correct?

Page 14. “Early horizontal transmission is crucial for parasite invasion, but is outpaced by vertical transmission once the parasite becomes established.” This is a really interesting result! What is the mechanism driving this temporal pattern? Is it just a function of higher infection prevalence later in the season, or is there a change in larval recruitment dynamics? Could using uninfected first instar larvae overestimate environmental transmission rates? I would think 1st instar caterpillars are likely to remain close to their leaf of origin (where the change of vertical transmission is high) and later instars (and likely more resistant) will roam farther, potentially contacting the spores through environmental transmission.

Page 16 bottom paragraph. This is a great point about seemingly minor transmission modes having large epidemiological effects. And it’s definitely a compelling reason for modeling the dynamics! We have found similar outcomes in our plant -disease system where seemingly trivial movement of spores by wind has a much stronger effect on dynamics than pollinator vectors.

Page 17, last paragraph. I wonder if adult transfer plays an even more important role in disease dynamics in migratory species, as it seems uninfected carriers would be more efficient at moving the parasite than infected individuals that are unlikely to make the journey.

Another point of clarification: The calculation of birth rates (b_s and b_i) in the supplemental material is averaging birth rates out across females and males, correct?

On figure S3a: could you add dashed lines to show the estimated parameter value used for the main text results?

Review form: Reviewer 2 (Jordan Elouise Jones)

Recommendation

Major revision is needed (please make suggestions in comments)

Scientific importance: Is the manuscript an original and important contribution to its field?

Good

General interest: Is the paper of sufficient general interest?

Excellent

Quality of the paper: Is the overall quality of the paper suitable?

Good

Is the length of the paper justified?

Yes

Should the paper be seen by a specialist statistical reviewer?

No

Do you have any concerns about statistical analyses in this paper? If so, please specify them explicitly in your report.

No

It is a condition of publication that authors make their supporting data, code and materials available - either as supplementary material or hosted in an external repository. Please rate, if applicable, the supporting data on the following criteria.

Is it accessible?

N/A

Is it clear?

N/A

Is it adequate?

N/A

Do you have any ethical concerns with this paper?

No

Comments to the Author

This study elegantly integrates field and modelling work to examine the dynamics of a virulent protozoan in its host, the monarch butterfly, by taking advantage of a recent recolonization event. The paper demonstrates how incorporating all three transmission routes (vertical, environmental and adult-adult transfer) can maintain a virulent protozoan at a high prevalence in a population. The paper shows how all three transmission routes, and the high virulence of the protozoan can lead to declines in host abundance and suggests that this could be contributing to the significant declines of the monarch butterfly in North America. Indeed, the work adds to the growing body of literature highlighting the importance of incorporating all possible transmission routes to be able to reliably predict infection dynamics of wildlife or human disease. Overall, the paper is very well-written and easily accessible to a wide-ranging audience.

However, I feel that there are many important and relevant studies, which appear to have been ignored in the background and discussion of this study, and I am not sure the major thrust - that high prevalence/high virulence is rare and unexplained. Indeed, it is known from a few well-studied invertebrate systems that virulent pathogens can maintain high prevalence in natural populations, and that there are particular circumstances that lead to this, in terms of frequency dependent transmission and vertical transmission.

a) In the case of insect STIs, frequency-dependent transmission alone can allow a highly virulent infection to increase to high levels. For example, in the ladybird, *Adalia bipunctata*, a sexually transmitted mite (*C. hippodamiae*) causes females to become sterile after infection. Despite this, the infection undergoes seasonal epidemics taking the mite to 95% prevalence each year (Webberley et al., 2004; Ryder et al 2014). The stabilization of this dynamic is understood in terms of phenology and seasonal reproduction/birth (Pastok et al 2016). A parallel process is observed, albeit with lower virulence, for the sexually transmitted *Parobia* mite infection in the Eucalypt beetle (Seeman and Nahrung, 2004, Nahrung and Clarke, 2007).

b) In the Koala/*Chlamydia* system, mixed modes of transmission (sexual and vertical - through feeding) are likely to be contributing to the high infection prevalence of *Chlamydia*, despite being highly virulent particularly under stress, which can impact host abundance.

c) For vertical transmission, bacteria with male-specific virulence (male-killing) can reach very

high prevalence with major impact, and these can be stable over time. The association between vertical transmission and low virulence applies only to the female lineage (see work on *Acraea* butterflies by Jiggins et al 2000, and on *Hypolimnys* butterflies by Dyson et al 2004).

These works do not detract from the very nice case study and analysis presented; however they do require the introduction/discussion to be refocussed to take into account the known case studies that have investigated high prevalence/high virulence infections.

Minor corrections:

Typing error: numbers instead of number? (page 5, line 2)

Decision letter (RSPB-2019-0343.R0)

11-Apr-2019

Dear Ms Majewska:

I am writing to inform you that your manuscript RSPB-2019-0343 entitled "Multiple transmission routes sustain high prevalence of a virulent parasite in a butterfly host" has, in its current form, been rejected for publication in Proceedings B.

This action has been taken on the advice of referees, who have recommended that substantial revisions are necessary. With this in mind we would be happy to consider a resubmission, provided the comments of the referees are fully addressed. However please note that this is not a provisional acceptance.

Sincerely,

Proceedings B
mailto: proceedingsb@royalsociety.org

Associate Editor

Board Member: 1

Comments to Author:

I have now received referee reports and the Associate Editor's comments on your paper. As you can see they are positive about your work but raise a number of issues that need to be addressed before we can make the final decision. For me although firmly wedded in a single important system, the question - the role of multiple transmission modes in prevalence and disease impact - is an important general one that is not well studied. That said as one referee points out there is a literature in press that you need to address. The second referee also raises important points that need to be addresses.

Reviewer(s)' Comments to Author:

Referee: 1

Comments to the Author(s)

The authors used a combination of new field studies and published data to quantify three different transmission modes of a virulent parasite onto a population of non-migrating monarch butterflies. They developed a compartmental model, parametrized with field data, to estimate the impact of these three transmission modes on disease dynamics. They found that all three modes were needed to explain the high level of disease prevalence observed in the field, and that the contribution of these modes differed over time.

I really enjoyed reading this paper. Transmission is at the heart of disease dynamics and yet, we still know much less about transmission in natural systems than virulence. The study makes the important point the seemingly trivial transmission routes (adult -transfer) can nevertheless play an critical role in disease establishment and spread. It also nicely illustrates the point that multiple transmission routes can lead to high sustained prevalence, even for a highly virulent disease. I found the manuscript to be well written overall, but I have a few questions and points of clarification.

Point of clarification: It is mentioned that there are 6 plots. Are these close enough that the dynamics are the same? (Is the model for the whole patch?)

Out of curiosity, how does one mark adult monarchs?

How were the milkweed leaves sampled? Were these always the top leaves? What proportion of the leaves had larvae on them to begin with?

Fig 1 and table 4: The Density-dependent per capita larval mortality rate (u_1) and mortality rate of infected adults (u_I) look very similar.

Page 11 first paragraph: "The stage of collection, and the interaction between stage and month, were not significant predictors of infection prevalence" -what does 'stage of collection' mean?

Does it mean adult vs larvae, or larval stages (because larvae v. adult looks like a big effect)?

The line "Adult transfer of spores most commonly occurs through mating of uninfected females to infected males (at rate \square)." on page 9 seems surprising, because sex-specific differences in transmission have not been mentioned yet. I would suggest citing the source in the text (I know it is in table S4) or mentioning this bit of natural history in the introduction.

It is a bit unclear to me how the 'composite virulence measure' (described on pages 9 & 10) is incorporated into the model. It looks like the the individual components of virulence are

incorporated into eq 2, 3 (which makes sense). Is this composite measure just for the analysis shown in Fig 4? If so, you might tweak the wording in the methods to clarify.

Related: page 13, top paragraph. Point of clarification: monarch population size is determined through the model output, and virulence is that composite measure reference earlier, correct?

Page 14. "Early horizontal transmission is crucial for parasite invasion, but is outpaced by vertical transmission once the parasite becomes established." This is a really interesting result! What is the mechanism driving this temporal pattern? Is it just a function of higher infection prevalence later in the season, or is there a change in larval recruitment dynamics? Could using uninfected first instar larvae overestimate environmental transmission rates? I would think 1st instar caterpillars are likely to remain close to their leaf of origin (where the change of vertical transmission is high) and later instars (and likely more resistant) will roam farther, potentially contacting the spores through environmental transmission.

Page 16 bottom paragraph. This is a great point about seemingly minor transmission modes having large epidemiological effects. And it's definitely a compelling reason for modeling the dynamics! We have found similar outcomes in our plant-disease system where seemingly trivial movement of spores by wind has a much stronger effect on dynamics than pollinator vectors.

Page 17, last paragraph. I wonder if adult transfer plays an even more important role in disease dynamics in migratory species, as it seems uninfected carriers would be more efficient at moving the parasite than infected individuals that are unlikely to make the journey.

Another point of clarification: The calculation of birth rates (b_s and b_i) in the supplemental material is averaging birth rates out across females and males, correct?

On figure S3a: could you add dashed lines to show the estimated parameter value used for the main text results?

Referee: 2

Comments to the Author(s)

This study elegantly integrates field and modelling work to examine the dynamics of a virulent protozoan in its host, the monarch butterfly, by taking advantage of a recent recolonization event. The paper demonstrates how incorporating all three transmission routes (vertical, environmental and adult-adult transfer) can maintain a virulent protozoan at a high prevalence in a population. The paper shows how all three transmission routes, and the high virulence of the protozoan can lead to declines in host abundance and suggests that this could be contributing to the significant declines of the monarch butterfly in North America. Indeed, the work adds to the growing body of literature highlighting the importance of incorporating all possible transmission routes to be able to reliably predict infection dynamics of wildlife or human disease. Overall, the paper is very well-written and easily accessible to a wide-ranging audience.

However, I feel that there are many important and relevant studies, which appear to have been ignored in the background and discussion of this study, and I am not sure the major thrust - that high prevalence/high virulence is rare and unexplained. Indeed, it is known from a few well-studied invertebrate systems that virulent pathogens can maintain high prevalence in natural populations, and that there are particular circumstances that lead to this, in terms of frequency dependent transmission and vertical transmission.

a) In the case of insect STIs, frequency-dependent transmission alone can allow a highly virulent infection to increase to high levels. For example, in the ladybird, *Adalia bipunctata*, a sexually transmitted mite (*C. hippodamiae*) causes females to become sterile after infection. Despite this, the infection undergoes seasonal epidemics taking the mite to 95% prevalence each year (Webberley et al., 2004; Ryder et al 2014). The stabilization of this dynamic is understood in terms of phenology and seasonal reproduction/birth (Pastok et al 2016). A parallel process is observed,

albeit with lower virulence, for the sexually transmitted *Parobia* mite infection in the Eucalypt beetle (Seeman and Nahrung, 2004, Nahrung and Clarke, 2007).

b) In the Koala/*Chlamydia* system, mixed modes of transmission (sexual and vertical – through feeding) are likely to be contributing to the high infection prevalence of *Chlamydia*, despite being highly virulent particularly under stress, which can impact host abundance.

c) For vertical transmission, bacteria with male-specific virulence (male-killing) can reach very high prevalence with major impact, and these can be stable over time. The association between vertical transmission and low virulence applies only to the female lineage (see work on *Acraea* butterflies by Jiggins et al 2000, and on *Hypolimnys* butterflies by Dyson et al 2004).

These works do not detract from the very nice case study and analysis presented; however they do require the introduction/discussion to be refocussed to take into account the known case studies that have investigated high prevalence/high virulence infections.

Minor corrections:

Typing error: numbers instead of number? (page 5, line 2)

RSPB-2019-1630.R0

Review form: Reviewer 2

Recommendation

Accept with minor revision (please list in comments)

Scientific importance: Is the manuscript an original and important contribution to its field?

Good

General interest: Is the paper of sufficient general interest?

Excellent

Quality of the paper: Is the overall quality of the paper suitable?

Excellent

Is the length of the paper justified?

Yes

Should the paper be seen by a specialist statistical reviewer?

No

Do you have any concerns about statistical analyses in this paper? If so, please specify them explicitly in your report.

No

It is a condition of publication that authors make their supporting data, code and materials available - either as supplementary material or hosted in an external repository. Please rate, if applicable, the supporting data on the following criteria.

Is it accessible?

N/A

Is it clear?

N/A

Is it adequate?

N/A

Do you have any ethical concerns with this paper?

No

Comments to the Author

The authors have done a substantial job in improving the manuscript entitled 'Multiple transmission routes sustain high prevalence of a virulent parasite in a butterfly host'. Most importantly, the introduction of the study has been improved considerably. Previously, I felt that the main thrust of the study – that high prevalence/high virulence was rare and unexplained, was not entirely true. However, the introduction has now been refocussed and takes into account important and relevant invertebrate studies which have placed the study into a much wider context. The paper is still very well written and was a very interesting and enjoyable read. Following the resubmission I only have two minor comments that should be addressed.

- 1) Line 67: Overlapping generations are indeed required for the maintenance of STIs in the case studies described. However, overlapping generations are not required for the maintenance of Wolbachia in Hypolimnas system as it is maternally transmitted.
- 2) Line 68: Virulence may be adaptive to parasite transmission – keeping a healthy mating host, aiding maternal transmission.

Decision letter (RSPB-2019-1630.R0)

06-Aug-2019

Dear Ms Majewska

I am pleased to inform you that your manuscript RSPB-2019-1630 entitled "Multiple transmission routes sustain high prevalence of a virulent parasite in a butterfly host" has been accepted for publication in Proceedings B.

The referee and Associate Editor have recommended publication, but have also suggested some minor revisions to your manuscript. Therefore, I invite you to respond to their comments and revise your manuscript. Because the schedule for publication is very tight, it is a condition of publication that you submit the revised version of your manuscript within 7 days. If you do not think you will be able to meet this date please let us know.

[http://datadryad.org/submit?journalID=RSPB&manu=\(Document not available\)](http://datadryad.org/submit?journalID=RSPB&manu=(Document+not+available)) which will take you to your unique entry in the Dryad repository. If you have already submitted your data to dryad you can make any necessary revisions to your dataset by following the above link. Please see <https://royalsociety.org/journals/ethics-policies/data-sharing-mining/> for more details.

Sincerely,

Professor Loeske Kruuk
mailto:proceedingsb@royalsociety.org

Associate Editor
Board Member
Comments to Author:

Thank you for revising your paper so well. The referee has a few minor suggestions to make that you should consider. I really like the paper.

I really like the paper. This time reading it through, however, I found the use of 'virulent pathogen' a little confusing - I know what you mean but you are using it as a short hand for "highly virulent", "very virulent", "substantially virulent" etc right - since if defined as the harm that a parasite causes its host there is no such thing as an "avirulent parasite" - parasites have to harm their hosts right -

Annoyingly, I wouldn't change the title, but I would make this point clearly by line 25 - "highly virulent" - line 29 with a "very" or something similar - line 44 I would say highly - line line 52 - I would say low again rather than 'non' -- I hope this is not too annoying - I think the short hand works most of the time but it should be made clear to non experts with a scattering of highly and avoiding avirulent or non-virulent. I'm sure you had this in earlier versions, but in this final read through I really felt it needs to be clear.

Overall I loved the paper - a great use of modelling in a well understood important system - thank you for the opportunity to review it. .

Reviewer(s)' Comments to Author:

Referee: 2

Comments to the Author(s).

The authors have done a substantial job in improving the manuscript entitled 'Multiple transmission routes sustain high prevalence of a virulent parasite in a butterfly host'. Most

importantly, the introduction of the study has been improved considerably. Previously, I felt that the main thrust of the study – that high prevalence/high virulence was rare and unexplained, was not entirely true. However, the introduction has now been refocussed and takes into account important and relevant invertebrate studies which have placed the study into a much wider context. The paper is still very well written and was a very interesting and enjoyable read. Following the resubmission I only have two minor comments that should be addressed.

1) Line 67: Overlapping generations are indeed required for the maintenance of STIs in the case studies described. However, overlapping generations are not required for the maintenance of *Wolbachia* in *Hypolimnas* system as it is maternally transmitted.

2) Line 68: Virulence may be adaptive to parasite transmission – keeping a healthy mating host, aiding maternal transmission.

Decision letter (RSPB-2019-1630.R1)

14-Aug-2019

Dear Ms Majewska

I am pleased to inform you that your manuscript entitled "Multiple transmission routes sustain high prevalence of a virulent parasite in a butterfly host" has been accepted for publication in *Proceedings B*.

Open Access

Paper charges

Sincerely,

Editor, Proceedings B
mailto: proceedingsb@royalsociety.org